# FilterNet: A Many-to-Many Deep Learning Architecture for Time Series Classification

**DOI:** 10.3390/s20092498

**Published:** 2020-04-28

**Authors:** Robert D. Chambers, Nathanael C. Yoder

**Affiliations:** Pet Insight Project, Kinship, 1355 Market St #210, San Francisco, CA 94103, USA; rdchambers@whistle.com

**Keywords:** activity recognition, time series classification, neural, networks, deep learning, machine learning, CNNs, LSTMs, many-to-many

## Abstract

In this paper, we present and benchmark FilterNet, a flexible deep learning architecture for time series classification tasks, such as activity recognition via multichannel sensor data. It adapts popular convolutional neural network (CNN) and long short-term memory (LSTM) motifs which have excelled in activity recognition benchmarks, implementing them in a many-to-many architecture to markedly improve frame-by-frame accuracy, event segmentation accuracy, model size, and computational efficiency. We propose several model variants, evaluate them alongside other published models using the Opportunity benchmark dataset, demonstrate the effect of model ensembling and of altering key parameters, and quantify the quality of the models’ segmentation of discrete events. We also offer recommendations for use and suggest potential model extensions. FilterNet advances the state of the art in all measured accuracy and speed metrics when applied to the benchmarked dataset, and it can be extensively customized for other applications.

## 1. Introduction

Time series classification is a challenging problem in numerous fields [1], including finance [2], cyber security [3], electronic health record analysis [4], acoustic scene classification [5], and electroencephalogram (EEG)-based brain computer interfaces [6], and it is a central challenge in the field of activity recognition [7]. Numerous time series classification algorithms have been proposed [8,9], and the diversity of time series classification problems is evident in dataset repositories, such as the UCR Time Series Archive [10] or the UCI Machine Learning Repository [11]. While the time series classification algorithm described in this work is applicable to many of these domains, it was developed as an activity recognition algorithm, and we present and benchmark it here primarily in that context.

Activity recognition (AR) is the classification of a subject’s moment-to-moment activities, typically using an input time series acquired via electronic sensors. AR is used in applications as diverse as smart homes [12,13], gesture recognition [14], computer control interfaces [15], health monitoring [16], and home behavior analysis [17].

Body-worn sensors are among the most common data sources, and of these, accelerometers are among the most common types of sensor, partially due to their low cost and low energy consumption. Sometimes accelerometers are accompanied by gyroscopes and magnetometers, as in an inertial measurement unit. Other sensors may include microphones, pressure sensors, and various object-mounted sensors that detect object displacement.

Classification can be performed with a variety of algorithms, ranging from traditional K-nearest-neighbors (KNN) and support vector machine (SVM) approaches that operate on hand-crafted features, to the more recent deep learning-based approaches that represent the current state of the art [7].

### 1.1. Motivation

The model described in this work was developed as part of the Pet Insight Project [18], a large pet health study that involves using activity recognition algorithms to measure health-related canine behavior. The project aimed to provide a resource for studying and improving pet health by combining electronic medical records and owner-provided survey data with a high-resolution longitudinal record of each behaviors and activity levels. It intended to include data from over 100,000 dogs over 2–3 years. Behavior recognition was performed on 3-axis 50 Hz accelerometer time series, recorded via a collar-mounted Whistle activity monitor [19].

Assembling this dataset required highly accurate behavior recognition with a very low false positive rate. The classification problem was made more challenging by differences in dog behavior and morphology, collar tightness, and even the positioning of the device on the collar and the natural movement of the collar around and along the dog’s neck. Predictions must be relatively high-frequency (3 Hz or more) in order to resolve short-duration events such as barking and shaking, and it was helpful if the output time series could be easily segmented into events. Finally, cost-effectively processing hundreds of thousands of device-years of data with such high input and output frequencies required algorithms to be extremely computationally efficient.

To our knowledge, besides the model described here, there has been no algorithm published that achieves the level of prediction accuracy that we require, while being cost effective to operate at scale. Furthermore, as wearable devices become more ubiquitous, and as machine learning algorithms are increasingly deployed to the edge for continuous real-time monitoring, we expect that the need for accurate and extremely computationally efficient activity recognition and event segmentation algorithms will only intensify.

In a future report, we plan to describe the application of the FilterNet model to dog behavior detection in the Pet Insight Project. However, for clarity and reproducibility, we limited the scope of this article to publicly available benchmark data.

### 1.2. Traditional Approaches

Until the relatively recent advent of deep-learning AR approaches, traditional AR methodologies [7] involved: (1) acquiring appropriate time series; (2) calculating hand-crafted statistical and spectral feature vectors over finite temporal windows; (3) training models such as KNN, SVM, naïve Bayes, or random forests, which mapped those feature vectors to activity predictions; and (4) evaluating those models on new or held-out time series data to infer activities. Over the years, a large variety of classifiers have been proposed and benchmarked, culminating in ensemble-based methods such as the Collective of Transformation-based Ensembles (COTE) [8] and its hierarchical voting variant HIVE-COTE [20]. However, the improved accuracy of the ensemble methods comes at the cost of a greatly increased computational burden [21].

### 1.3. Deep Learning Approaches

In deep learning approaches, the traditional machine learning model is replaced with a so-called deep learning or neural network model with many layers [7]. These models can achieve very high classification accuracy without the need for hand-crafted features [22,23].

Many deep learning AR models are composed at least partially of convolutional neural network (CNN) components. While traditional neural networks usually train a learned weight for every input-output pair, CNNs instead convolve trainable fixed-length kernels (filters) along their inputs. Often, pooling and striding are used to reduce the size of the CNN’s output in the dimensions that the convolution is performed, reducing computational cost and making overtraining less likely [24]. Excellent AR results have been obtained using 1-D CNNs to process fixed-length time series segments produced with sliding windows [25,26]. These models may also include pooling and striding, and they typically run in a many-to-one configuration, often concatenating the outputs of the final CNN layer and using a fully-connected layer to produce a single class prediction at each time step. Some methodologies transform time-series signals into 2D matrices that can be classified using image classification models, typically using a sliding-window approach [27,28,29].

While 1-D CNNs convolve fixed-length kernels along an input signal, recurrent neural networks (RNNs) instead process each time step sequentially, so that an RNN layer’s final output is a function of every preceding timestep. However, basic RNNs suffer from various shortcomings [30], and most recent work employs more complex variants. Perhaps the most successful RNN variant is the long short-term memory (LSTM) model [31], which extends the basic RNN with a memory cell and several control gates in order to better model time dependencies in long sequences [32]. LSTMs are typically unidirectional (i.e., they process the time series in the order it was recorded), but if an entire input sequence is available then two parallel LSTMs can be evaluated in opposite directions (e.g., forwards and backwards in time) and their results can be concatenated, forming a bidirectional LSTM (bi-LSTM) that can model temporal dependencies in both directions.

Ordóñez and Roggen combine CNNs and LSTMs in their DeepConvLSTM model [33], which advanced the state of the art considerably as measured by the Opportunity [34] and Skoda [35] benchmarks. DeepConvLSTM consists of a stack of four unstrided CNN layers followed by two LSTM layers and a softmax classifier. The input signals to the CNNs are not padded, so even though the layers are unstrided, each CNN layer shortens the time series by several samples. The LSTM layers are unidirectional, and so the softmax classification corresponding to the final LSTM output is used in training and evaluation, as well as in reassembling the output time series from the sliding window segments. DeepConvLSTM operates primarily in a many-to-one configuration.

Hammerla et. al. [36] describe and compare several classifiers, including dense, CNN, and LSTM variants. The LSTM variants include LSTM-F, a unidirectional LSTM model that operates in a many-to-one mode, and LSTM-S (unidirectional) and base LSTM (b-LSTM)-S (bidirectional), which can operate in a many-to-many mode, generating an output class prediction output for every input timestep, and obviating the need for sliding-window segmentation. These models were trained with innovative techniques to minimize overtraining, and the LSTM variants did not include CNN preprocessing layers.

LSTMs have proved to be well suited to AR tasks, and several of the highest performing models employ them in one form or another [37,38,39].

### 1.4. Our Approach

Our FilterNet approach incorporates ideas from several existing models, but combines them in an arrangement that is potentially more accurate and often much faster. Its characteristics include:**Many-to-many**. Our approach can process entire input signals at once (see Figure 1), and does not require sliding, fixed-length windows.**Striding/downsampling**. If appropriate, our approach can use striding to reduce the number of samples it outputs. This can both improve computational efficiency and enable subsequent layers to model dynamics over longer time ranges.**Multi-scale**. Like the well-known U-Net image segmentation model [40], FilterNet can downsample beyond its output frequency to model even longer-range temporal dynamics.

This work describes a prototypical FilterNet architecture, enumerates several variants with different architectural and performance characteristics, benchmarks those variants and compares them to results from the literature, explores the effect of adding or removing model components and of altering model parameters, and offers recommendations for use and customization.

## 2. FilterNet

We chose the name FilterNet in order to emphasize a key and distinguishing property of this class of architecture—namely, that like the finite impulse response (FIR) filters common in signal processing [41], a FilterNet model can be applied to time series of arbitrary length, and it will infer an output time series of length proportional to the input length. This is true for FilterNet because it is true for all of its constituent building blocks—1D CNNs, LSTMs, pooling layers, interpolation layers, batch normalization layers, and so on. As a many-to-many (i.e., sequence-to-sequence) model, the architecture of a FilterNet model is not tied to the length of its input, and a larger time series length or sliding window length does not require a larger FilterNet model.

### 2.1. FilterNet Layer Modules (FLMs)

FilterNet models are composed primarily of a stack of parameterized modules, which we will refer to here as FilterNet layer modules (FLMs). They are meant to be easily combined into signal-processing stacks, to be easily tweaked and re-configured, and to train efficiently. They are also coverage-preserving; that is, even though the input and output of an FLM may differ in sequence length due to a stride ratio, the time period that the input and output cover will be identical. The basic components of an FLM are shown in Figure 2a. For brevity, we describe FLMs with the notation:(1)$$FLM_{type}\left(w_{\mathrm{out}},\;s=1,\;k=5,p_{\mathrm{drop}}=0.1,\;b_{BN}=\mathrm{True}\right)$$
where type is the type of the primary trainable sub-layer (‘*cnn*’ for a 1-D CNN or ‘*lstm*’ for a bi-directional LSTM); $w_{\mathrm{out}}$ is the number of output channels (the number of filters for a *cnn* or the dimensionality of the hidden state for an *lstm*); s is a stride ratio (default 1); k is the kernel length (only for CNNs, default 5), and $p_{\mathrm{drop}}$ is the dropout probability (default 0.1). If s>1, then a 1-D average-pooling with stride s and pooling kernel length *s* reduces the output length by a factor of s.

Each FLM consists of an (optional) dropout layer which randomly drops out input channels during training with probability $p_{\mathrm{drop}}$; either a 1D CNN with *ReLU* activation or a bidirectional LSTM layer with *tanh* activation (depending on FLM type); a 1D average-pooling layer which pools and strides the output of the CNN or LSTM layer whenever s≠0 (we may refer to these simply as *strided* layers, but these always include a matching pooling step so that all CNN or LSTM output samples are represented in the FLM output); and a 1D batch normalization layer. The dropout layer and/or batch normalization layer(s) serve to regularize the network and improve training dynamics [42].

All CNN layers were configured to zero-pad their input by $\mathrm{ceil}\left(\frac{k-1}{2}\right)$, so that their input and output signal lengths were equal. Consequently, each FLM maps an input tensor $X_{\mathrm{in}}$ of size $\left[w_{\mathrm{in}},L_{\mathrm{in}}\;\right]$ to an output tensor $X_{\mathrm{in}}$ of size $\left[w_{\mathrm{out}},L_{\mathrm{out}}=\;L_{\mathrm{in}}/s\right]$.

Other modifications can sometimes be useful, such as gated recurrent unit (GRU) layers, ‘grouping’ of CNN filters, and different strategies for pooling, striding, and dilation; however, these are avoided here for clarity and brevity.

### 2.2. Component Architecture

The potential configurations of a stack of FLMs are virtually limitless. Therefore, we present a prototypical FilterNet architecture in Figure 2b that we have found to be performant and applicable to a wide range of use cases. Applications which require highly optimized performance may benefit from modifying this architecture.

This prototypical FilterNet architecture is composed of six optional components, each of which can be parameterized or simply removed to adjust the network’s properties. These components are:
**(A) Full-Resolution CNN** (*s* = 1, *t* = cnn). *High-resolution processing.* Convolves CNN filters against the input signal without striding or pooling, in order to extract information at the finest available temporal resolution. This layer is computationally expensive because it is applied to the full-resolution input signal.**(B) Pooling Stack 1** (*s* > 1, *t* = cnn). *Downsamples from the input to the output frequency*. This stack of $n_{p1}$ CNN modules (each strided by *s*) downsamples the input signal by a total factor of $s^{n_{p1}}$. The output length of this stack determines the output stride ratio, $s_{\mathrm{out}}=s^{n_{p1}}$, and thus the output length of the network for a given input, $L_{out}=L_{in}/s^{n_{p1}}$.**(C) Pooling Stack 2** (*s* > 1, *t* = cnn). *Downsamples beyond the overall output frequency.* This stack of $n_{p_{2}}$ modules (again, each strided by *s*) further downsamples the output of the previous layer in order to capture slower temporal dynamics. To protect against overtraining, the width of each successive module is reduced by a factor of s so that $w_{i}=w_{p}s^{1-i}$ for $i=1,\ldots ,n_{p_{2}}$.**(D) Resampling Step**. *Matches output lengths.* In this step, every output of Pooling Stack 2 (C) is resampled in the temporal dimension via linear interpolation to match the network output length $L_{out}$. These outputs are concatenated with the final module output of Pooling Stack 1 (B). Without this step, the lengths of the outputs of (C) would not match the output length, and so they could not be processed together in the next layer. We have found that exposing each intermediate output of (C) in this manner, as opposed to only exposing the final output of (C), improves the model’s training dynamics and accuracy.**(E) Bottleneck Layer**. *Reduces channel number.* This module effectively reduces the width of the concatenated outputs from (D), reducing the number of learned weights needed in the recurrent stack (F). This bottleneck layer allows a large number of channels to be concatenated from (C) and (D) without resulting in overtraining or excessively slowing down the network. As a CNN with kernel length k=1, it is similar to a fully connected dense network applied independently at each time step.**(F) Recurrent Stack** (*s* = 1, *t* = lstm). *Temporal modeling*. This stack of $n_{l}$ recurrent LSTM modules provides additional modeling capacity, enables modeling of long-range temporal dynamics, and improves the output stability of the network.**(G) Output Module** (*s* = 1, *k* = 1, *t* = cnn). *Provides predictions for each output time step*. As in (E), this is implemented as a CNN with k=1, but in this case without a final batch normalization layer. The multi-class outputs demonstrated in this work use a *softmax* activation function.

Again, not all components are beneficial for every application, and each component can be independently reconfigured or removed to optimize the model’s properties.

### 2.3. FilterNet Variants

In order to demonstrate the effects of adding and removing components, we define several model variants in Table 1, ranging from simpler to more complex.

Of these, only the base LSTM (b-LSTM) does not downsample the network input. It is a stack of (usually one or two) $FLM_{LSTM}$ layers followed by an output module. It is most structurally similar to the b-LSTM-S architecture demonstrated by Hammerla et. al. [36]. This model would likely benefit from additional CNN layers, but we include it here for demonstration purposes.

The pooled CNN (p-CNN) is simply a stack of $FLM_{CNN}$ layers where one or more of the layers is strided, so that the output frequency is lower than the input frequency. This improves computational efficiency and increases the timescales that the network can model, relative to an unstrided CNN stack. The pooled CNN/LSTM (p-C/L) adds one or more recurrent layers that operate at the output frequency immediately before the output modules layer. The p-C/L architecture is in some ways similar to the DeepConvLSTM [33] architecture, although the striding and pooling features of p-C/L, and its many-to-many approach, lead to substantially different performance characteristics.

The multiscale CNN (ms-CNN) and multiscale CNN/LSTM (ms-C/L) variants modify the p-CNN and p-C/L variants by adding a second pooling stack and subsequent resampling and bottleneck layers. This progression from p-CNN to ms-C/L demonstrates the effect of increasing the variants’ ability to model long-range temporal interactions, both through additional layers of striding and pooling, and through recurrent LSTM layers.

## 3. Materials and Methods

In this work, we demonstrate and benchmark several FilterNet variants, and we compare the results to our implementation of the DeepConvLSTM model [33] and to select results from the literature.

### 3.1. Benchmark Dataset

While many quality datasets are available for benchmarking AR algorithms [43], we used the Opportunity Activity Recognition dataset to demonstrate FilterNet due to its wide usage as a benchmark dataset in the activity recognition field [7], its relatively large size (about 6 h of recordings), and its diverse array of sensors and labels (from which we can choose various subsets). Furthermore, like the real-world datasets for which FilterNet was developed, much of the Opportunity dataset consists of the null class—that is, regions without labeled behaviors. The full dataset was released publicly following its use in the Opportunity activity recognition challenge [34], and it can be freely downloaded online [44]. It consists of four subjects performing normal morning activities in a sensor-rich setting. Each subject was recorded performing a single practice (Drill) session of predefined and scripted activities and five sessions of activities of daily living (ADL) in an undefined order. The dataset is provided at a 30 Hz frequency.

We used the same train and test sets used in the Opportunity challenge. Specifically, we attempted to follow the data processing steps employed by Ordóñez and Roggen [33] as closely as possible, since other works have differed markedly in, e.g., their handling of missing values or their choice of validation and test sets [36,45]. Hammerla et al. specify a test set matching that in this paper, but the publicly available implementation uses a test set of subjects 2 and 3, runs 3 and 4 [36]. Also, Hammerla et. al. appear to skip samples with missing data, as opposed to interpolating the missing data. In all instances, we held out ADL sessions 4 and 5 for subjects 2 and 3 as a test set, and we did not use data from subject 4. The Opportunity challenge uses data from subject 4, but Ordóñez and Roggen do not [33]. Unless otherwise specified (e.g., n-fold ensemble models), we held out ADL session 3 for subjects 2 and 3 as a validation set, and train our models on the remaining ADL and Drill sessions.

Following Ordóñez and Roggen [33], we only used the 113 sensor channels used in the original challenge [34], which comprise the outputs from 7 inertial measurement units (IMU) with accelerometer, gyroscope, and magnetic sensors, and 12 Bluetooth accelerometers, and we used linear interpolation to fill in missing sensor data. However, instead of rescaling and clipping all channels to a [0,1] interval using a predefined scaling, we re-scaled all data to have zero mean and unit standard deviation according to the statistics of the training set. Where noted (e.g., multimodal fusion analysis in Section 4.4), we restricted our experiments to a subset of these 113 sensor channels.

We reproduced the ‘Task B’ challenge of inferring the occurrence of sporadic gestures such as using doors or kitchen appliances. This is an 18-class (including the null class) classification task.

### 3.2. Reference Architectures

Even the six model variants in Table 1 can vary widely in their implementation due to the number of layers in each component and the configuration (e.g., striding and number of output channels) in each layer. Therefore, we demonstrated and benchmarked a specific reference architecture for each variant. We kept model parameters consistent between the variants to facilitate comparison. The specific number, sizes, and configurations of each layer for these reference architectures are shown in Table 2, where the layers are constructed and arranged as shown in Figure 2.

For clarity, we also present the ms-C/L reference architectures in more detail in Table 3. This was the largest of our reference architectures, and the others are composed primarily of subsets of the ms-C/L layers (although changes in layer input sizes may affect the exact number of trainable parameters). We defined the region of influence (ROI) for a layer as the maximum number of input samples that can influence the calculation of a single output sample. This region is increased by larger kernels, by larger stride ratios, and by additional layers. It represents an upper limit on the timescales that an architecture is capable of modeling. Note that it was only calculated for CNN-type layers, since the region of influence of bi-directional LSTMs was the entire input. The ${\mathrm{ROI}}_{i}$ for a $FLM_{CNN}\left(s_{i},\;k_{i}\right)$ layer *i* that is preceded in a stack only by other $FLM_{CNN}$ layers can be calculated by:(2)$${\mathrm{ROI}}_{i}{=\mathrm{ROI}}_{i-1}+\left(k_{i}-1\right)\sum_{j=1}^{i}s_{j}$$

We also implemented a version of DeepConvLSTM [33] using the publicly available source code [46] as a reference. Our libraries and training methodology vary from the original and we were unable to match the original’s performance, but we report the runtime performance results here to quantify approximate speed differences.

### 3.3. Software and Hardware Specifications

We implemented the models in Python using *PyTorch* v1.0.1 [47] and the 2019.10 release of the Anaconda Python distribution (64-bit, Python 3.7.5). We used the ward-metrics v0.9.5 library [48] for calculating and plotting event metrics according to Ward et. al. [49]. We trained and evaluated our models on a p2.xlarge instance on Amazon Web Services [50] with 4 vCPUs (Intel Xeon E5-2686 v4), 61 GB RAM, and a NVIDIA Tesla k80 GPU with 12 Gb RAM, running Ubuntu 16.04. The source code to reproduce the models, results, and visualizations in this paper are publicly available on GitHub at https://github.com/WhistleLabs/FilterNet.

### 3.4. Inference Windowing

FilterNet is a many-to-many model. Consequently, it can process signals of any length. However, due to memory, efficiency, and sometimes latency constraints, it is typically helpful to: (a) divide long input signals into segments using a sliding window of a moderate fixed length and with some segment overlap; (b) process the segments in batches sized appropriately for available memory; and (c) reconstruct the corresponding output signal from the processed segments.

Typically, classification accuracy suffers near the start and end of each segment due to edge effects. Overlap between segments allows these edge regions to be removed without creating gaps in the output signal. Alternatively, to prevent signal discontinuities, segments can be averaged using a weighted window that de-emphasizes the edge regions, as in this work.

Unless otherwise noted, we used a sliding window length of 512 samples with 50% overlap and a Hanning window for weighted averaging of overlap regions.

### 3.5. Performance Metrics

We calculated validation and test set performance using both sample-based and event-based metrics. Sample-based metrics were aggregated across all class predictions, and were not affected by the order of the predictions. Event-based metrics were calculated after the output is segmented into discrete events, and they are strongly affected by the order of the predictions.

We calculated sample-based precision, recall, and F1 scores for each output class, including the null class. Then, for comparison with the various metrics reported in the literature, we summarized overall model performance as either a mean F1 score averaged across the non-null classes ($F_{1m})$, or as a weighted F1 score ($F_{1w})$, across all classes, where each class is weighted according to its sample proportion in the ground-truth label set. When required for comparison to other works, we also reported a non-null weighted F1 score ($F_{1w,\;nn})$ which ignores the null class.

For event-based metrics, we followed the recommendations of Ward et. al. [49]. We also defined an event F1 metric ($F_{1e})$ in order to summarize these extensive metrics. We calculated $F_{1e}$ in terms of true positives (TP), false positives (FP), and false negatives (FN):(3)$$F_{1e}=2\frac{\mathrm{Precision}\;\cdot \;\mathrm{Recall}}{\mathrm{Precision}+\mathrm{Recall}}=2\frac{\frac{\mathrm{TP}}{\mathrm{TP}+\mathrm{FP}}\cdot \frac{\mathrm{TP}}{\mathrm{TP}+\mathrm{FN}}}{\frac{\mathrm{TP}}{\mathrm{TP}+\mathrm{FP}}+\frac{\mathrm{TP}}{\mathrm{TP}+\mathrm{FN}}}$$

Here, TP events are correct (C) events as defined by Ward, while FN events are incorrect actual events (D, F, FM, and M in Ward et. al.), and FP events are incorrect returned events (M’, FM’, F’, and I’ in Ward et. al.). To calculate the overall $F_{1e}$, we simply sum the TP, FP, and FN counts across all classes. This score is not weighted by event length, so long events have the same influence as short events.

We also report training (GPU) speed as seconds/epoch, and inference speed (both GPU and CPU-only) as input samples processed per second, measured using our GPU-enabled compute instance.

### 3.6. Hyperparameter Search

We conducted a hyperparameter search using the Ray v0.7.6 distributed application framework and its included Tune scalable hyperparameter tuning library. We ran hundreds of trials to evaluate the effect of training and architectural parameters on prediction accuracy. However, due to relatively high run-to-run variance caused by the small number of subjects and runs in the Opportunity dataset, and potentially due to the use of LSTMs in several models (which we find can decrease repeatability run-to-run), we were unable to achieve finely tuned hyperparameter choices with this approach.

### 3.7. Model Training

We trained the models on GPUs using the parameters in Table 4, unless otherwise noted. We divided the training and validation sets into segments using a sliding window. We chose window lengths that are integer multiples of all models’ output stride ratios in order to minimize implementation complexity. Because window length varied in some of our experiments, we adjusted the batch size to hold the total number of input samples in a batch approximately constant.

We found that the validation loss was too noisy to be used as an early stopping metric, due to the small number of subjects and runs in the validation set. Consequently, we used a custom stopping metric that is more robust and which penalizes oscillations in performance, so that stopping does not occur until model performance is relatively stable between epochs. Specifically, we defined a smoothed validation metric as the exponentially weighted moving average (with a half-life of 3 epochs) of $l_{v}/F_{1w,v}$ where $l_{v}$ is the validation loss and $F_{1w,v}$ is the weighted F1 score of the validation set, calculated after each training epoch. This metric decreases as the loss and/or the F1 score improve. We also calculate an instability metric as the standard deviation of the past five $l_{v}/F_{1w,v}$ values. We summed these metrics to yield a checkpoint metric. The model is checkpointed whenever the checkpoint metric reaches a new minimum, and training is stopped after patience epochs without checkpointing.

A representative training run is shown in Figure 3 to illustrate this process. We found that this approach allowed us to train models consistently across architectures using small validation sets and without excessive tuning.

### 3.8. Ensembling

Where noted, we performed *n*-fold ensembling by: (a) combining the training and validation sets into a single contiguous set; (b) dividing that set into *n* disjoint folds of contiguous samples; (c) training *n* independent models where the *i*_th_ model uses the *i*_th_ fold for validation and the remaining *n*-*1* folds for training; and (d) ensembling the *n* models together during inference by simply averaging their logit outputs before the softmax function was applied. Performance of the overall ensemble was still measured on the same test set used in other experiments; only the train and validation sets varied. For efficiency, the evaluation and ensembling of the *n* models was performed using a single computation graph in PyTorch.

## 4. Results

### 4.1. Model Performance

We summarize the performance of the reference FilterNet models on the Opportunity dataset gestures task in Table 5. We also include three variants of the ms-C/L architecture: a 4-fold ensemble of the ms-C/L architecture, and scaled versions in which the $w_{out}$ values were scaled by ½ or by 2×. Finally, in order to demonstrate key differences with prior work, we include the runtime performance of our reimplementation of the DeepConvLSTM architecture.

The 4-fold ms-C/L model is substantially more accurate than the simpler variants by all of our accuracy measures, and particularly in terms of event-based metrics. Indeed, we have found that 3-to-5-fold ms-C/L ensembles perform well on many tasks and datasets, especially if inference speed and model size are not of critical importance.

Figure 4 shows ground-truth labels and model predictions for approximately 150 s during the first run in the standard Opportunity test set, for several models. It demonstrates qualitatively why FilterNet models score highly in event-based performance metrics. FilterNet models—especially fully-featured ones such as the ms-C/L architecture—produce far fewer short, spurious events. This reduces the false positive count, while also preventing splitting of otherwise correct events. For instance, in the region of interest shown, the event-based $F_{1e}$ metric increases from 0.66 in Figure 4d to 1.0 in Figure 4b, while the sample-by-sample $F_{1w}$ metric increases only from 0.84 to 0.90. In applications where event counts are quantified, or where actions are taken in response to discrete events, high event segmentation performance is critical. The event segmentation performance that FilterNet achieves can obviate the need for further event processing and downselection. Often, FilterNet outputs can produce excellent event segmentation results by simply thresholding the output signal at *p* = 0.5 and enumerating the contiguous regions, as we have done in this work.

Figure 5 shows results for the same models as in Figure 4, but calculated for the entire test set, and broken out into more detail as recommended by Ward e. al. [49]. The event summary diagrams compare the ground truth labels (actual events) to model predictions (detected events). Correct events (C) do not necessarily indicate exact agreement, but simply that there is a 1:1 correspondence between actual and detected events. The event summary diagrams depict the number of actual events that are missed (D; deleted) or multiply detected (F; fragmented), as well as the detected fragments (F’; fragmenting) and any spurious detections (I’; insertions).

It is evident that the lower performing models Figure 5c,d suffered primarily from low precision; the b-LSTM implementation detected 117 out of 204 events correctly, but it generated 448 spurious or fragmented events. The ms-CNN model Figure 5b demonstrates the effect of adding additional strided layers to the p-CNN model, which increased the model’s ROI from 61 to 765 samples, meaning that the ms-CNN model can model dynamics occurring over a 12× longer region of influence. The 4× ms-C/L ensemble Figure 5a was improved further by addition of an LSTM layer, and by making it difficult for a single model to register a spurious event without agreement from the other ensembled models.

### 4.2. Model Ensembling

The effects of model ensembling on accuracy (sample-by-sample $F_{1w}$ and event-based $F_{1e}$), as well as the inference rate of the ensemble, are plotted in Figure 6. As described earlier, these models are trained on n−1 folds, with the remaining fold used for validation. The 2-fold models, therefore, have validation sets equal in size to their test sets, and the train and validation sets are simply swapped in the two sub-models. The higher-*n* models have a more traditional train-validation split (approx. 67%:33%, 75%:25%, and 80%:20% for the 3-, 4-, and 5-fold ensembles, respectively) and this is likely why the average sub-model performance increases with *n* in the plotted range.

It is notable that the event-based metrics Figure 6b benefit more from ensembling than the sample-by-sample metrics Figure 6a (as measured by the difference between the ensemble and sub-model metrics), likely because the ensembling helps to suppress short, uncertain events unless the majority of sub-models are in agreement.

It is unclear whether ensembling would be as beneficial on a larger dataset with, e.g., more subjects and experimental runs so that each fold would have a more representative and consistent collection of behaviors.

### 4.3. Window Length Effects

Figure 7 demonstrates the effect of changing the sliding window length used in the inference step. As described above, although FilterNet is able to process time series of arbitrary length, efficiency and memory constraints necessitate windowing in most applications, and some overlap is necessary to reduce edge effects in those windows. For simplicity, this work used windows with 50% overlap, weighted with a Hanning window to de-emphasize edges and prevent the introduction of discontinuities where windows meet. Batch size was 100 windows in this figure.

While model accuracy is monotonically non-decreasing with window length, the inference rate reaches a maximum for LSTM-containing models where the efficiencies of constructing and reassembling longer segments, and the efficiencies of some parallel execution on the GPUs, balance the inefficient sequential execution of the LSTM layer on GPUs. While this balance can vary, we found that windows of 256 to 2048 samples tended to perform well. On CPUs, these effects were less prominent due to less parallelization, although very short windows still can exhibit overhead. The efficiency drawbacks of executing LSTMs on GPUs can be substantially mitigated by using a GPU-optimized LSTM implementation (e.g., cuDNN) as we have done in this work, and by using an architecture with a large output-to-input stride ratio so that the input sequence to the LSTM layer is shorter.

We note that if FilterNet models do not include any LSTM layers (e.g., the p-CNN and ms-CNN variants, above), then they have a finite ROI, and edge effects are only possible within ROI/2 of the window ends. Consequently, windows only need to overlap by approximately ROI/2 input samples, and the windows can simply be concatenated after discarding half of each overlapped region. If this windowing strategy is used, then the efficiency benefit of longer windows is even more pronounced, and especially considering the excellent parallelizability of CNNs, it may be advisable to use a batch size of 1 and to simply use the longest window length possible given system memory constraints.

In almost all cases that we have explored, GPUs achieved far greater inference rates than CPUs and are more cost efficient at typical cloud computing prices. However, if models are small (few trainable parameters), or are primarily LSTM-based, or if time series are especially short, then CPU execution may be preferred.

### 4.4. Performance Using Fewer Sensor Channels

FilterNet models are especially well-suited to datasets with relatively few sensors. Table 6 reproduces the multimodal fusion analysis of [33], wherein models are trained and evaluated on the same train, validation, and test sets, but with different subsets of sensor outputs ranging from 15 to 113 channels. We held models’ architecture parameters constant where possible, but note that the number of trainable parameters in the first model layer will vary when the number of input channels changes. We compared the DeepConvLSTM results reported in [33] with a p-CNN model (which had a relatively small ROI of 61 samples and moderate performance on this dataset), an ms-C/L model (which had large/unbounded ROI and high performance on this dataset), and a 4-fold ensemble of ms-C/L models. We extend this analysis in Figure 8, plotting both $F_{1w,\;nn}$ and the event-based $F_{1e}$ across the same set of sensor subsets, for all 5 FilterNet reference architectures and a 4-fold ms-C/L ensemble.

As expected, the 4-fold ms-C/L ensemble was the most accurate model, especially according to event-based metrics. It is interesting to note the consistent performance of the ms-C/L, ms-CNN, and p-C/L models, even with fewer sensors. All of these models have long or unbounded ROIs, which may help them to compensate for the missing sensor channels. We also note that the FilterNet models often perform best on the 45-sensor subset, indicating possible overtraining or a suboptimal architecture for the full complement of sensors.

For all models in Table 6, accelerometers appeared to be more useful than gyroscopes, but models using both performed best. It is unclear if the usefulness of the magnetic sensors would persist in real-world scenarios where subjects may be oriented in any direction while performing tasks.

### 4.5. Comparison to Published Results

Table 7 compares the benchmark results described in this paper to similar results from the literature. Performance comparisons on this dataset are imperfect, since details such as the exact test set used and the handling of missing values are not always identical or even explicit, and the performance metrics employed vary widely. The source code used to train and evaluate the models is often unavailable, so key implementation details are unknown. Furthermore, since the test set is publicly available, it is often not clear whether the training parameters were tuned to maximize test set performance. Finally, performance is sometimes reported using the ‘best’ model (as measured by test set performance) instead of as an unbiased average of independent replicates. In other cases, the train and test sets are shuffled in a way that prevents fair comparison.

Regardless, FilterNet advances the state of the art on all of the performance metrics in Table 7 even though several of the reported metrics select (or may select) the best test-set performance post-hoc from multiple runs. It is surprising that the ms-CNN model, which does not contain a recurrent layer, exhibits such high performance. In [39] Guan and Plötz demonstrate excellent performance using an ensemble of twenty carefully trained LSTM learners which, like FilterNet, use a many-to-many architecture. It is likely that an ensembling strategy such as that used in [39], combined with the pooling and multiscale features exhibited in this work, could improve performance even further.

## 5. Discussion

### 5.1. Recommendations for Use

The preceding results attempt to illustrate the performance improvements possible with a fully-featured FilterNet architecture. However, compared to simple FilterNet architectures, full-featured ones are typically slower to train, slower to run, exhibit more variance from run-to-run (complicating model exploration and hyperparameter tuning), and are far more prone to overtraining on small datasets.

For most situations, we recommend beginning exploration with a simple, small p-CNN architecture with the largest output:input stride ratio that can fully resolve the shortest events of interest. For instance, if a dataset has a 50 Hz input sampling rate and output events as short as 1 s, then a 16:1 output:input stride ratio will resolve each event with at least three output samples. This might suggest, e.g., four pooling layers with stride ratios of 2, or two pooling layers with stride ratios of 4, and so on, depending on the desired model capacity. The resulting architecture can be tuned relatively quickly because the model should be fast and consistent from run to run.

If the p-CNN model exhibits adequate performance, then it could be useful to try a similar ms-CNN architecture, wherein the second pooling stack (component C) is long enough that the longest dynamics of interest are within its region of influence. If overtraining is evident, it can be especially helpful to lower $w_{out}$ for the layers in component C, since their large ROI causes the dataset to effectively appear smaller with respect to its ability to prevent overtraining. Once the ms-CNN model is tuned, and if it does not exhibit problematic overtraining, then adding one or more recurrent LSTM layers (component F) may substantially improve accuracy and/or event segmentation performance. However, not all datasets exhibit improved performance with these layers, and the simpler ms-CNN architecture can be much faster and more consistent run-to-run. Finally, once an accurate and properly optimized model is found, it can be useful to simply create a 3-to-5-fold ensemble using the same training and architecture parameters.

### 5.2. Other Modifications

The approach used in FilterNet is far more flexible than we are able to fully demonstrate in this work. For instance, it is possible to use the network to simultaneously calculate multiple independent outputs. For instance, the same network can simultaneously predict both a quickly-varying ‘behavior’ and a slowly-varying ‘posture’, such as the ‘gestures’ and ‘locomotion’ labels in the Opportunity dataset. The loss functions for the multiple outputs can be simply added together, and the network can be trained on both simultaneously. This enables some degree of automatic transfer learning between the two label sets, but also presents challenges in optimally training the network.

FilterNet can also be easily adapted to multi-label classification problems and regression problems by simply changing the output type(s) (e.g., changing the final activation function from softmax to sigmoid or linear) and the loss functions (e.g., from cross-entropy to binary cross-entropy or mean squared error), and it may be possible to combine these as independent outputs in the same model. In our experience, these changes are relatively straightforward but usually require re-tuning of the training parameters.

Furthermore, the approach outlined in this work can be easily extended with new layer types, although we have found the layers presented above to form a strong and widely applicable basis for modeling many sensor time series. However, GRUs or other RNNs sometimes outperform LSTMs, especially when computational resources are limited. Also, networks with extremely high stride ratios may require additional measures such as dilation of the CNN layers or longer CNN kernels to prevent gaps where input data or intermediate activations are ignored. Finally, it may be possible to improve the layer modules for certain applications by using skip connections or even a heterogeneous inception-like architecture. We also note that the basic FilterNet approach can be extended to real-time or streaming applications by, for instance, using only CNNs or by replacing bidirectional LSTMs with unidirectional LSTMs. Because FilterNet’s probability outputs represent its classification confidence, such real-time or streaming networks could be used in early classification systems to minimize classification latency [54].

We expect that these real-time or streaming FilterNet implementations are efficient enough for deployment on mobile or embedded platforms, especially those platforms with a suitable AI coprocessor.

## 6. Conclusions

FilterNet arose from our need for activity recognition models that could match or exceed the state-of-the-art accuracy, while running at much lower cost. We also aimed to lessen the need for additional event segmentation or event down-selection steps. We found that the FilterNet approach described in this article substantially improves upon existing methods in all of these areas, setting new records in $F_{1w}$, $F_{1m}$, and $F_{1m,nn}$ for the Opportunity dataset. While speed comparisons are difficult without freely available reference implementations of existing models in compatible frameworks, FilterNet’s many-to-many approach, combined with its abilities to achieve very high stride ratios and to model relatively long-term dynamics without a strict need for LSTMs, give practitioners important levers with which to minimize computation costs. Finally, as shown above, FilterNet architectures are able to largely obviate the need for custom event segmentation steps, especially when they include multi-scale and/or LSTM components.

## Figures and Tables

**Figure 1 sensors-20-02498-f001:**
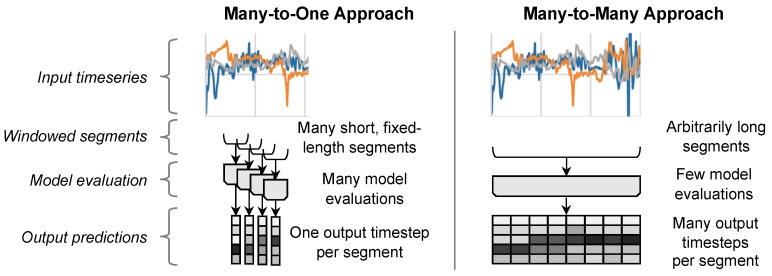
In a typical many-to-one approach (**left**) an input is first divided into fixed-length overlapping windows, then a model processes each window individually, generating a class prediction for each one, and finally the predictions are concatenated into an output time series. In a many-to-many approach (**right**), the entire output time series is generated with a single model evaluation.

**Figure 2 sensors-20-02498-f002:**
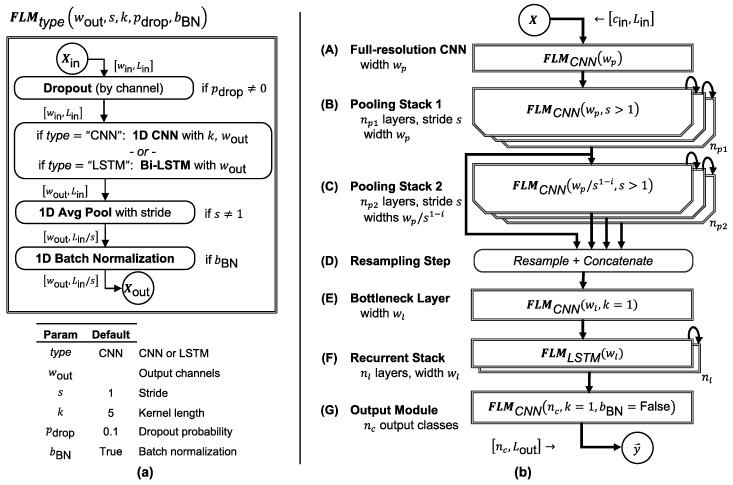
FilterNet architecture. (**a**) Each FilterNet model is composed primarily of one or more stacks of FilterNet layer modules (FLMs), which are parameterized and constructed as shown (see text for elaboration). (**b**) In the prototypical FilterNet component architecture, FLMs are grouped into components that can be parameterized and combined to implement time series classifiers of varying speed and complexity, tuned to the problem at hand.

**Figure 3 sensors-20-02498-f003:**
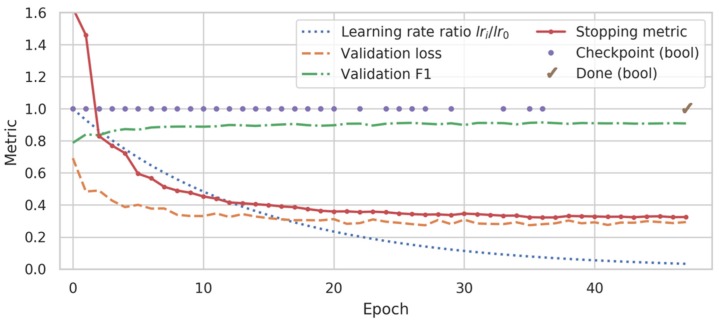
Representative training history for a ms-C/L model. While the validation loss oscillated and had near-global minima at epochs 27, 35, and 41, the custom stopping metric (see text) adjusted more predictably to a minimum at epoch 36. Training was stopped at epoch 47 and the model from epoch 36 was restored and used for subsequent inference.

**Figure 4 sensors-20-02498-f004:**
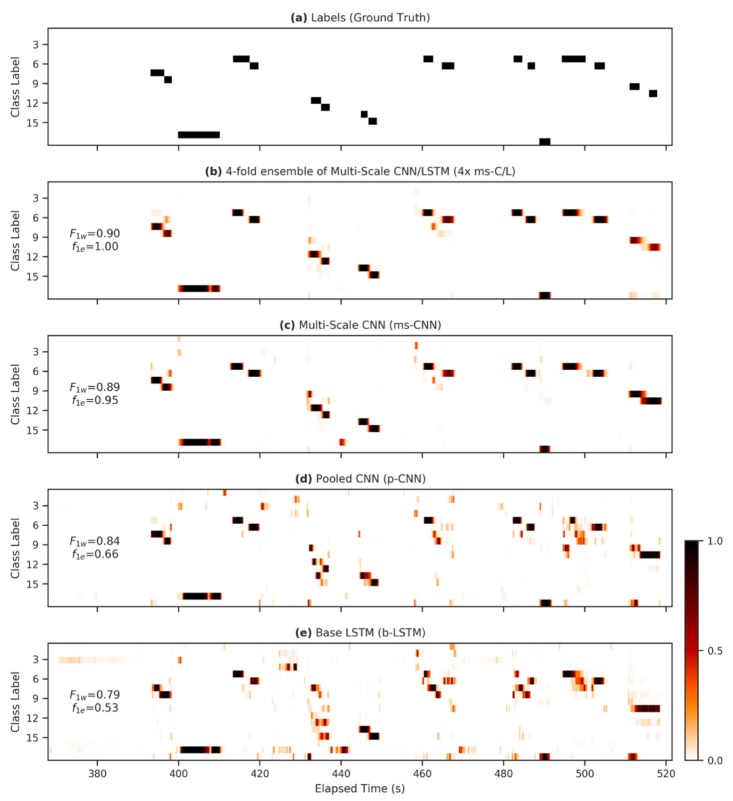
Heatmap demonstrating differences between model outputs. (**a**) The ground-truth labels for 150 s during the first run in the standard Opportunity test set; (**b–e**) alongside the corresponding predictions for the 17 non-null behavior classes for various FilterNet architectures. Panes are annotated with weighted F1 scores ($F_{1w}$ and the event-based $F_{1e}$) calculated over the plotted region.

**Figure 5 sensors-20-02498-f005:**
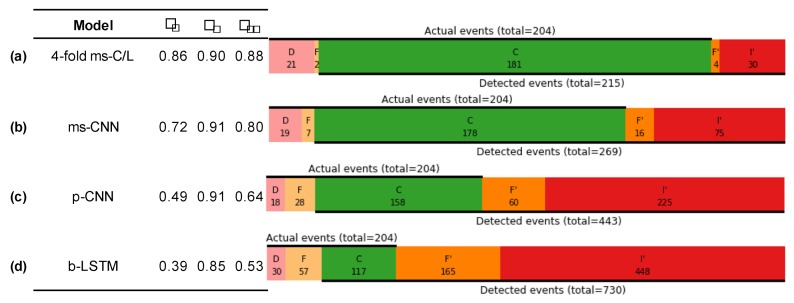
Event-based metrics. Performance metrics for several classifiers, including event-based precision $P_{e}$, recall $R_{e}$, and F_1_ score $F_{1e}$, alongside event summary diagrams, each for a single representative run.

**Figure 6 sensors-20-02498-f006:**
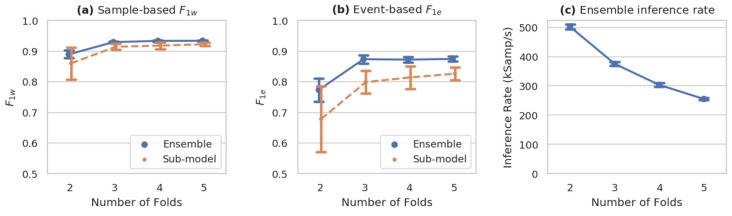
Performance of *n*-fold ensembled ms-C/L models. Both sample-based (**a**) and event-based (**b**) weighted F1 metrics improve substantially with the number of ensembled models, plateauing between 3–5 folds, while inference rate (pane **c**) decreases. (mean ± sd, *n* = 10).

**Figure 7 sensors-20-02498-f007:**
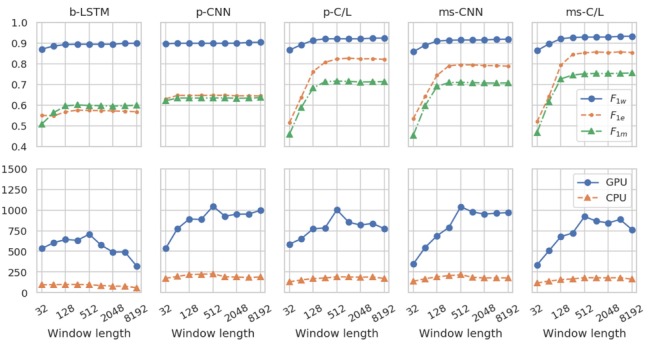
Effects of inference window length. For all models, accuracy metrics (top panes) improve as inference window length increases, especially when models incorporate multi-scale or LSTM architectural features. For LSTM-containing models, the inference rate (bottom panes) falls off for long windows because their calculation across the time dimension cannot be fully parallelized. (*n* = 5 each).

**Figure 8 sensors-20-02498-f008:**
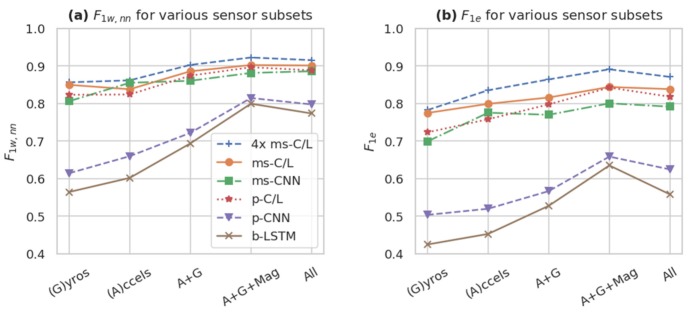
Performance of different Opportunity sensor subsets (mean of *n* = 5 runs) according to (**a**) the sample-by-sample $F_{1w,\;nn}$ and (**b**) the event-based F1 $F_{1e}$. Using larger sensor subsets, including gyroscopes (G), accelerometers (A), and the magnetic (Mag) components of the inertial measurement units, as well as all 113 standard sensors channels (All), tended to improve performance metrics. The best models (e.g., 4× ms-C/L, ms-C/L, ms-CNN, and p-C/L) maintain relatively high performance even with fewer sensor channels.

**Table 1 sensors-20-02498-t001:** Several FilterNet variants (and their abbreviated names). The variants range from simpler to more complex, using different combinations of components A-G. CNN: convolutional neural network; LSTM: long short-term memory.

Variant	Components						
	A	B	C	D	E	F	G
Base LSTM (b-LSTM) ^1^	-	-	-	-	-	✓	✓
Pooled CNN (p-CNN)	✓	✓	-	-	-	-	✓
Pooled CNN/LSTM (p-C/L)	✓	✓	-	-	-	✓	✓
Multi-Scale CNN (ms-CNN)	✓	✓	✓	✓	✓	-	✓
Multi-Scale CNN/LSTM (ms-C/L)	✓	✓	✓	✓	✓	✓	✓

^1^ Base LSTM has a 1:1 output stride ratio, unlike the other variants.

**Table 2 sensors-20-02498-t002:** Configuration of the reference architectures used in this article. Each configuration represents one of the named FilterNet variants.

Component	b-LSTM	p-CNN	p-C/L	ms-CNN	ms-C/L
A—Full-Res CNN	-	$L_{\mathrm{CNN}}\left(100\right)$			
B—Pooling Stack 1		$L_{\mathrm{CNN}}\left(100,\;s=2\right)$ $L_{\mathrm{CNN}}\left(100,\;s=2\right)$ $L_{\mathrm{CNN}}\left(100,\;s=2\right)\to i_{1}$			
C—Pooling Stack 2	-			$L_{\mathrm{CNN}}\left(100,\;s=2\right)\to i_{2}$ $L_{\mathrm{CNN}}\left(50,\;s=2\right)\to i_{3}$ $L_{\mathrm{CNN}}\left(25,\;s=2\right)\to i_{4}$ $L_{\mathrm{CNN}}\left(13\right)\to i_{5}$	
D—Resampling Step	-			195 output channels ^1^	
E—Bottleneck Layer	-			$L_{\mathrm{CNN}}\left(100,\;k=1\right)$	
F—Recurrent Layers	$L_{\mathrm{LSTM}}\left(100\right)$	-	$L_{\mathrm{LSTM}}\left(100\right)$	-	$L_{\mathrm{LSTM}}\left(100\right)$
G—Output Module	$L_{\mathrm{CNN}}\left(18,\;k=1\right)$				

^1^ Resamples intermediates $i_{2,\ldots ,5}$ to each have $\mathrm{len}\left(i_{1}\right)$. Concatenate with $i_{1}$ for 195 output channels with matching lengths.

**Table 3 sensors-20-02498-t003:** Layer-by-layer configuration and description of the multi-scale CNN/LSTM (ms-C/L), the most complex reference architecture.

Component	Type	*W_in_*	*W_out_*	s	k	**Params** ^1^	**Output Stride Ratio** ^2^	**ROI** ^3^
in	input	113 ^4^						1
A	$FLM_{\mathrm{CNN}}$	113 ^4^	100	1	5	56,700 ^4^	1	5
B	$FLM_{\mathrm{CNN}}$	100	100	2	5	50,200	2	13
B	$FLM_{\mathrm{CNN}}$	100	100	2	5	50,200	4	29
B	$FLM_{\mathrm{CNN}}$	100	100	2	5	50,200	8	61
C	$FLM_{\mathrm{CNN}}$	100	50	2	5	25,100	16	125
C	$FLM_{\mathrm{CNN}}$	50	25	2	5	6300	32	253
C	$FLM_{\mathrm{CNN}}$	25	13	2	5	1651	64	509
C	$FLM_{\mathrm{CNN}}$	13	7	1	5	469	64	765
D	*resample*	195	195					765
E	$FLM_{\mathrm{CNN}}$	195	100	1	1	19,700	8	765
F	$FLM_{\mathrm{LSTM}}$	100	100	1		80,500	8	*all*
G	$FLM_{\mathrm{CNN}}$	100	18	1	1	1818	8	*all*
out	output		18					*all*

^1^ Number of trainable parameters, including those for bias and batch normalization. ^2^ Ratio of layer output frequency to system input frequency. ^3^ Region of influence; see text for detail. ^4^ Varies when sensor subsets are used.

**Table 4 sensors-20-02498-t004:** Training parameters used in this study, along with an approximate recommended range for consideration in similar applications.

Parameter	Value	Recommended Range
Max epochs	100	50–150
Initial learning rate	0.001	0.0005–0.005
Samples per batch	5,000	2,500–10,000
Training window step	16	8–64
Optimizer	Adam	Adam, RMSProp
Weight decay	0.0001	0–0.001
Patience	10	5–15
Learning rate decay	0.95	0.9–1.0
Window length	512	64–1024

**Table 5 sensors-20-02498-t005:** Model and results summary for the Opportunity dataset, the five FilterNet reference architectures, as well as three modifications of the ms-C/L architecture. The 4-fold ms-C/L variant exhibited the highest accuracy, while smaller and simpler variants were generally faster but less accurate. The best results are in bold.

		Architecture		Classification Metrics				Efficiency	
Model	*n* ^1^	Stride Ratio	Params (k)	$F_{1w}$	$F_{1m}$	$F_{1e}$	$F_{1w,\;nn}$	kSamp/s	Trains/epoch
	**FilterNet reference architectures**								
b-LSTM	9	1	87	0.895	0.595	0.566	0.787	865	15
p-CNN	9	8	209	0.900	0.638	0.646	0.803	1340	**2.0**
p-C/L	9	8	299	0.922	0.717	0.822	0.883	1160	4.0
ms-CNN	9	8	262	0.919	0.718	0.792	0.891	1140	3.5
ms-C/L	9	8	342	0.928	0.743	0.842	0.903	1060	5.1
	**Other variants**								
4-fold ms-C/L	10	8	1371	**0.933**	**0.755**	**0.872**	**0.918**	303	5.1 × 4
½ scale ms-C/L	9	8	100	0.921	0.699	0.815	0.880	**1350**	5.2
2x scale ms-C/L	9	8	1250	0.927	0.736	0.841	0.901	682	7.0
	**Non-FilterNet models**								
DeepConvLSTM reimplementation	20	12	3965	--	--	--	--	9	170

^1^ Results are mean of *n* repeats.

**Table 6 sensors-20-02498-t006:** Model mean F1 score without nulls ($F_{1w,\;nn}$) for different sensor subsets (*n* = 5 each). This table reproduces the multimodal fusion analysis of [33]. The ms-C/L and 4-fold ms-C/L models improve markedly upon reported DeepConvLSTM performance, especially with smaller sensor subsets. The best results are in bold.

	Gyros	Accels	Accels+ Gyros	Accels+ Gyros + Magnetic	Opportunity Sensors Set
# of sensor channels	15	15	30	45	113
DeepConvLSTM [33]	0.611	0.689	0.745	0.839	0.864
p-CNN	0.615	0.660	0.722	0.815	0.798
ms-C/L	0.850	0.838	0.886	0.903	0.901
4-fold ms-C/L	**0.857**	**0.862**	**0.903**	**0.923**	**0.916**

**Table 7 sensors-20-02498-t007:** Performance comparison alongside published models on the Opportunity dataset. FilterNet models perform set records in each metric. The best results are in bold.

Method/Model		$F_{1m}$	$F_{1w}$	$F_{1w,nn}$	*n*
DeepConvLSTM	[33]	0.672	0.915	0.866	?
LSTM-S	[36]	0.698	0.912		*best* of 128
		0.619			*median* of 128
b-LSTM-S	[36]	0.745	0.927		*best* of 128
		0.540			*median* of 128
LSTM Ensembles	[39]	0.726			*mean* of 30
Res-Bidir-LSTM	[51]		0.905		?
Asymmetric Residual Network	[52]		0.903		?
DeepConvLSTM + Attention	[53]	0.707			*mean*
FilterNet ms-CNN		0.718	0.919	0.891	*mean* of 9
FilterNet ms-C/L		0.743	0.928	0.903	*mean* of 9
FilterNet 4-fold ms-C/L		**0.755**	**0.933**	**0.918**	*mean* of 10

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
