# Peer review of "FilterNet: A Many-to-Many Deep Learning Architecture for Time Series Classification"

_sensors, 2020, doi:10.3390/s20092498_

Round 1

Reviewer 1 Report

This paper proposes a flexible deep learning architecture for time series
classification tasks (e.g., human activity recognition via multichannel sensor data) .The proposed scheme imported many-to-many,  striding / downsampling, and multi-scale characteristics, and received good performance in accuracy, model size, computational efficiency. My main concerns are as follows:

1. What kinds of sensors are used in this work? Is it possible for the proposed scheme to be deployed on resource limited mobile devices? Please give some statement.

2. Is there any impact on the performance due to different segmentation? Please give some statement or experimental evluation.

3. There is no introduction for the dataset. Please give the detailed illustration of “dataset”, or cite a reference.

4. There are a few typos.

5. There are many "Error! Reference source not found." Them should be fixed.

6. On page 4 and 17, the BLANK-SPACE should be fixed.

7. It is suggested to refer to the following state-of-art references on human activity recongnition:

-Learning Aided User Identification Using Smartphone Sensors for Smart Homes. IEEE Internet of Things Journal. (2019)

-A framework for collaborative computing and multi-sensor data fusion in body sensor networks. Information Fusion. (2015)

-Imaging and Fusing Time Series for Wearable Sensors based Human Activity Recognition. Information Fusion. (2020)

Author Response

Thank you for reviewing our submitted article, FilterNet: A many-to-many deep learning architecture for time series classification. We genuinely appreciate you taking the time to read the article and to help us improve it. We have modified the article as requested, and we respond to several specific points below.

Please note that we have attached a PDF copy of the submitted article to supplement the version that Sensors provides, to be sure that MS Word incompatibilities do not cause further trouble.

> 1. What kinds of sensors are used in this work? Is it possible for the proposed scheme to be deployed on resource limited mobile devices? Please give some statement.

The article includes the following section describing the sensors (lines 274-280)

Following Ordóñez and Roggen [33], we use only the 113 sensor channels used in the original challenge [34], which comprise the outputs from 7 inertial measurement units (IMU) with accelerometer, gyroscope, and magnetic sensors, and 12 Bluetooth accelerometers, and we use linear interpolation to fill in missing sensor data. However, instead of rescaling and clipping all channels to a [0,1] interval using a predefined scaling, we re-scale all data to have zero mean and unit standard deviation according to the statistics of the training set. Where noted (e.g., multimodal fusion analysis in Section 4.4), we restrict our experiments to a subset of these 113 sensor channels.

The reference to the Opportunity dataset, which contains further detail, is also included:

  1. Chavarriaga, R.; Sagha, H.; Calatroni, A.; Digumarti, S.T.; Tröster, G.; Millán, J. del R.; Roggen, D. The Opportunity challenge: A benchmark database for on-body sensor-based activity recognition. Pattern Recognit. Lett. 2013, 34, 2033–2042.

Our description is of a similar level of detail to the original DeepConvLSTM paper, also published in this journal.
Regarding mobile devices, we do expect that these real-time or streaming FilterNet implementations, in particular, will be efficient enough and well suited for deployment on mobile or embedded platforms, especially those platforms with a suitable AI coprocessor. We have added such a statement on lines 612-614

> 2. Is there any impact on the performance due to different segmentation? Please give some statement or experimental evluation.

Regarding sliding-window segmentation: this certainly is an issue, and we document the effects of sliding window length (the primary parameter) on three different F1 metrics and for five FilterNet variants across 9 window lengths on both CPUs and GPUs,  in Figure 7. We also discuss alternative windowing strategies and the possibility of largely avoiding sliding window segmentation.

Regarding segmentation of output events, any change to the event segmentation would certainly impact the results. We have chosen to use the simplest and least controversial event segmentation strategy that we are aware of, which is to simply segment the output at 50% confidence threshold and to group adjacent output samples that are above or below that threshold. We argue that the ability to achieve good results with such a simple strategy is one of the strengths of our approach. The papers we reference use a variety of approaches to tailor segmentation results to specific applications, and reimplementing them is beyond the scope of this work.

If we are misunderstanding the meaning of the question, please tell us.

> 3. There is no introduction for the dataset. Please give the detailed illustration of “dataset”, or cite a reference.

We introduce the dataset in Section 3.1, “Benchmark Dataset” (lines 255-282), and we include a reference (ref 34) to the article describing the dataset. This section is a 4-paragraph description of the Opportunity benchmark dataset that we use, including the reasons we chose it; a reference to the article introducing it; a reference to where it could be downloaded; a description of the activities included in it; a detailed description of what train and test data we use and why; and a description of which sensor channels we use and why, alongside references to other papers that use this dataset upon which we base our approach.

If there is something missing from this description, we are happy to edit it. We have attempted to describe the dataset in a level of detail that is similar to the key papers that we cite.

> 4. There are a few typos.

We have corrected several typos, and would be grateful to know of any others.

> 5. There are many "Error! Reference source not found." Them should be fixed.

We sincerely regret these formatting issues. The downloaded article does not contain these errors on our systems. We are aware that MS Word and Zotero can exhibit these issues, especially when interoperating between different versions (we use, and we apologize for any issues.

The submitted PDF version also did not contain any of these issues, and should be more platform-independent. We attach a PDF version of the updated article to this response.

We do our best to abide by MDPI's author guidelines and to use the software that they recommend, and we look forward to working with the editorial office to fix any problems. 

> 6. On page 4 and 17, the BLANK-SPACE should be fixed.

We have attempted to adhere as closely as possible to MDPI's suggested template for this submission. It does include some blank spaces between sections, subsections, and figures, as required. We have inserted some page breaks in order to prevent captions or section headings from breaking in unexpected places. It is possible that the issues are due to MS Word incompatibilities, and we are curious whether the blank space exists in the PDF version.

Regardless, we will be sure to work with MDPI to ensure that the final version is properly formatted.

In response to this comment, we have removed page breaks from the new version in case they were causing these problems. Consequently, some section headings or captions may be orphaned. Again, referring to the attached PDF version may be helpful.

> 7. It is suggested to refer to the following state-of-art references on human activity recongnition:

> -Learning Aided User Identification Using Smartphone Sensors for Smart Homes. IEEE Internet of Things Journal. (2019)

> -A framework for collaborative computing and multi-sensor data fusion in body sensor networks. Information Fusion. (2015)

> -Imaging and Fusing Time Series for Wearable Sensors based Human Activity Recognition. Information Fusion. (2020)

We have incorporated the references as [13], [9], and [28], respectively.

Reviewer 2 Report

The idea is intersting. However, the paper seems to be prepared in hurry. There are 15 "Error! Reference source not found"!! Please fix these, along with a careful tailoring of the paper. 

In literature review, there are some interesting state-of-the arts time series classification algorithms missing. For example, the authors didnt mention anything about 2D classification of time series, which seems is getting many attentions. 

- Implementation of Deep Neural Networks to Classify EEG Signals using Gramian
Angular Summation Field for Epilepsy Diagnosis, Abelino Jimenez & Bhiksha Raj, 2020.

- Classification of Time-Series Images Using Deep Convolutional Neural Networks, N Hatami, Y Gavet, J Debayle, ICMV, 2017.

Another interesting discussion would be to see if the proposed method is adaptive with the idea of "early classification/detection". The are many literature on this e.g. Classifiers with a reject option for early time-series classification, N Hatami, C Chira. 2013.

Another important point is lack for comparisons with state of the art algorithms for each dataset. Also comparisons with 2D methods would enrich the paper. 

Author Response

Thank you for reviewing our submitted article, FilterNet: A many-to-many deep learning architecture for time series classification. We genuinely appreciate you taking the time to read the article and to help us improve it. We have modified the article as requested, and we respond to several specific points below.

Please note that we have attached a PDF copy of the revised article to supplement the version that Sensors provides, to be sure that MS Word incompatibilities do not cause further trouble.

> The idea is intersting. However, the paper seems to be prepared in hurry. There are 15 "Error! Reference source not found"!! Please fix these, along with a careful tailoring of the paper. 

We are deeply sorry that the reviewers experienced these problems. We were extremely careful to try to avoid these problems, and the downloaded article does not contain these errors on our systems. However, we are aware that MS Word and Zotero can exhibit these issues, especially when interoperating between different versions 

The submitted PDF version did not contain any of these issues, and should be more platform-independent.

We do our best to abide by MDPI's author guidelines, and we look forward to working with the editorial office to fix any problems. However, beyond ensuring that the MS Word doc works properly on the authors' systems, we are not aware of a good solution other than using the PDF version.

We are attaching a PDF of the new, revised version, to this response. Like the originally submitted PDF, it does not exhibit these issues.

> In literature review, there are some interesting state-of-the arts time series classification algorithms missing. For example, the authors didnt mention anything about 2D classification of time series, which seems is getting many attentions. 

> - Implementation of Deep Neural Networks to Classify EEG Signals using Gramian

> Angular Summation Field for Epilepsy Diagnosis, Abelino Jimenez & Bhiksha Raj, 2020.

> - Classification of Time-Series Images Using Deep Convolutional Neural Networks, N Hatami, Y Gavet, J Debayle, ICMV, 2017.

Thank you - we have incorporated these references as [29] and [27].

> Another interesting discussion would be to see if the proposed method is adaptive with the idea of "early classification/detection". The are many literature on this e.g. Classifiers with a reject option for early time-series classification, N Hatami, C Chira. 2013.

Thank you - we have incorporated this reference as [54], and we have added a statement that, insomuch as FilterNet models produce useful probabilistic outputs, they should be adaptable to an early detection scheme. It is an interesting potential modification, and we would be excited to see FilterNet used in such a capacity.

> Another important point is lack for comparisons with state of the art algorithms for each dataset. Also comparisons with 2D methods would enrich the paper. 

In Table 6, we reproduce the 'multimodal fusion analysis' of a state-of-the-art publication, and compare our results to the published results in 5 experimental conditions and across three model variants.

In Table 7, we compare the results of three FilterNet variants to the state-of-the art results from six publications in the literature, across three common figures of merit, using the Opportunity benchmark dataset.

In Section 4.5, “Comparison to published results” (lines 539-555), we discuss the comparison of our results to state-of-the-art results in detail.

Reviewer 3 Report

Review of “FilterNet: A many-to-many deep learning architecture for time series classification”

The paper describes a deep learning framework for time series classification via multichannel sensor data. Although the authors claim that it can be used for many different sources of time series data, the paper focuses on the application as activity recognition algorithm, which makes it a well suited topic for the sensors journal since “Signal processing, data fusion and deep learning in sensor systems” is listed as part of its scope.

What makes this paper offering an interesting angle is that the recorded acceleration sensors are supposed to be worn by dogs instead of humans. The authors provide detailed insight into the challenges and methods resulting from this form of application. However, the framework is used on a common activity recognition dataset and this form of application is not mentioned again. The relating results would be an interesting addition.

The architecture of the FilterNet components are described in detail and a schematic is provided to visualise the structure. The components can be arranged in different variants, which are illustrated by table 1, which shows the used constellations. Tables 2 and 3 describe individual structures and layers in detail, but lack a caption explaining the content. However, due to the detailed description of used methods, one should still be able to reproduce the used network architectures.

The network variants are benchmarked against the commonly known DeepConvLSTM model by Ordóñez and Roggen, although the used model is an implementation with different libraries and training methods, not achieving the same performance as the original. The authors could offer a more detailed descriptions about the differences between their baseline implementation and the original. For the benchmark the Opportunity dataset is used, which is also a commonly accepted standard. The authors claim to have achieved a better performance than all other published models on the Opportunity dataset while reducing computation cost. This is a significant achievement and justifies the conclusions drawn in the paper.

Recommended Changes

Figure 1,3,6,7, and 8 are using the default pyplot colours, a sequential colourmap would provide better readability for colour-blind readers or if the paper is printed in greyscale colours. The heatmap in figure 4 is intuitive but could still use a legend to establish which values are high and which are low. Other plots have sufficiently detailed legends.

Most tables could use a independent captions instead of relying solely on the text for description.

Author Response

Thank you for reviewing our submitted article, FilterNet: A many-to-many deep learning architecture for time series classification. We genuinely appreciate you taking the time to read the article and to help us improve it. We have modified the article as requested, and we respond to several specific points below.

Please note that we have attached a PDF copy of the submitted article to supplement the version that Sensors provides, since several reviewers ran into trouble with what we presume are MS Word incompatibilities.

> What makes this paper offering an interesting angle is that the recorded acceleration sensors are supposed to be worn by dogs instead of humans. The authors provide detailed insight into the challenges and methods resulting from this form of application. However, the framework is used on a common activity recognition dataset and this form of application is not mentioned again. The relating results would be an interesting addition.

Again, thank you for your detailed reading of the paper. We agree that the real-world results of applying the model to dogs is interesting and we are excited to share our results.

To do so, we plan on publishing those results in a follow-up paper. It is very difficult to be able to describe FilterNet in a reproducible way using our proprietary dog-behavior dataset, due to legal issues. In order to keep the paper as open and reproducible as possible, so that we could maximize our contribution to the technical community, we decided to split this paper off from the dog-specific paper.

We hope that the openness of this paper, and the reproducible code that we are hosting on Github, will make the real-world results paper more useful.

We now mention the planned follow-up paper on lines 69-71.

> The architecture of the FilterNet components are described in detail and a schematic is provided to visualise the structure. The components can be arranged in different variants, which are illustrated by table 1, which shows the used constellations. Tables 2 and 3 describe individual structures and layers in detail, but lack a caption explaining the content. However, due to the detailed description of used methods, one should still be able to reproduce the used network architectures.

As suggested, we have expanded our table captions. We appreciate this feedback, and we feel that the paper is easier to digest with these changes.

We have also now received clearance from our legal department to post the code on Github, which we hope makes the paper even easier to reproduce and build on.

We have added a link to our source code ( https://github.com/WhistleLabs/FilterNet )  as a foot note to line 319.

> The network variants are benchmarked against the commonly known DeepConvLSTM model by Ordóñez and Roggen, although the used model is an implementation with different libraries and training methods, not achieving the same performance as the original. The authors could offer a more detailed descriptions about the differences between their baseline implementation and the original. For the benchmark the Opportunity dataset is used, which is also a commonly accepted standard. The authors claim to have achieved a better performance than all other published models on the Opportunity dataset while reducing computation cost. This is a significant achievement and justifies the conclusions drawn in the paper.

We strongly wish that we were able to re-implement DeepConvLSTM in a more convincing way, and we put a great deal of time and effort into doing so. 

Unfortunately, since our results did not consistently match (only a few isolated runs matched the published performance), we didn’t feel that highlighting them offered much to the reader beyond what the original paper offered.

Furthermore, getting close to DeepConvLSTM’s published performance required a much different training approach, which made it difficult to run an even-handed comparison, and which made it very hard to succinctly describe the experimental conditions.

Ultimately, we decided it was better to focus on our results and to compare to DeepConvLSTM’s published results, so as not to misrepresent that work.

In that vein, we have further removed our DeepConvLSTM reimplementation results, except for the size and speed comparisons. (That is, we removed the DeepConvLSTM reimplementation from the F1 scores in Table 5; from the heat map in Figure 4; and from the Ward Metrics comparison in Table 5.)

We hope that these changes improve our article and avoid giving any false impressions of the performance of DeepConvLSTM. While we would like to have offered a more insightful comparison with DeepConvLSTM, it is most important to us not to misrepresent it.

> Figure 1,3,6,7, and 8 are using the default pyplot colours, a sequential colourmap would provide better readability for colour-blind readers or if the paper is printed in greyscale colours. The heatmap in figure 4 is intuitive but could still use a legend to establish which values are high and which are low. Other plots have sufficiently detailed legends.

We use the color scheme and plotting style that the Seaborn Python package recommends for scientific papers. We aim to use aesthetics that readers are familiar with and which are not distracting, but we agree that the grayscale and colorblind considerations are important. 

We feel that a sequential colormap would imply an ordering or preference to our data that we do not mean to imply, and we do not feel that any color scheme can effectively differentiate very many lines when printed in grayscale. 

Instead, we try to use differently shaped markers for each line. However, we failed to do this with Figure 3, and we have now added appropriate markers and line styles. We thank the reviewer for their attention to this detail. Figure 1 is for illustration only and the line colors are not meant to be important to its interpretation.

As requested, we have added a colormap to Figure 4 (we have tried to keep it compact so that the other details on this complex figure remain readable).

> Most tables could use a independent captions instead of relying solely on the text for description.

We appreciate this feedback. We have added more detailed captions to most tables, and we feel that it improves the paper’s readability.

Round 2

Reviewer 2 Report

The comments and suggestions from the 1st round of reviews are addressed. As far as I am concerned, the manuscript is good for publication.